# Genetic evolution of influenza viruses among selected countries in Latin America, 2017–2018

Juliana Almeida Leite[1], Paola Resende[2], Jenny Lara Araya[3], Gisela Badillo Barrera[4], Elsa Baumeister[5], Alfredo Bruno Caicedo[6], Leticia Coppola[7], Wyller Alencar de Mello[8], Domenica de Mora[6], Mirleide Cordeiro dos Santos[8], Rodrigo Fasce[9], Jorge Fernández[9], Natalia Goñi[7], Irma López Martínez[4], Jannet Otárola Mayhua[10], Fernando Motta[2], Maribel Carmen Huaringa Nuñez[10], Jenny Ojeda[11], María José Ortega[12], Erika Ospitia[13], Terezinha Maria de Paiva[14], Andrea Pontoriero[5], Hebleen Brenes Porras[3], Jose Alberto Diaz Quinonez[4,15], Viviana Ramas[7], Juliana Barbosa Ramírez[13], Katia Correa de Oliveira Santos[14], Marilda Mendonça Siqueira[2], Cynthia Vàzquez[12], Rakhee Palekar[1]*

1 Pan American Health Organization (PAHO/WHO), Washington, DC, United States of America,
2 Laboratorio de Virus Respiratorio, Fundação Oswaldo Cruz (Fiocruz), Rio de Janeiro, Rio de Janeiro, Brazil, 3 Instituto Costarricense de Investigación y Enseñanza en Nutrición y Salud (INCIENSA), Tres Ríos, Cartago, Costa Rica, 4 Instituto de Diagnóstico y Referencia Epidemiológicos (InDRE), Ciudad de México, Mexico, Mexico, 5 Instituto Nacional de Enfermedades Infecciosas—Administración Nacional de Laboratorios e Institutos de Salud (INEI-ANLIS) "Dr. Carlos G. Malbran", Buenos Aires, Argentina, 6 Instituto Nacional de Investigación en Salud Pública (INSPI), Guayaquil, Guayas, Ecuador, 7 Departamento de Laboratorio de Salud Publica (DLSP), Montevideo, Montevideo, Uruguay, 8 Instituto Evandro Chagas (IEC), Ananindeua, Para, Brazil, 9 Instituto de Salud Pública de Chile (ISPCH), Santiago, Santiago, Chile, 10 Instituto Nacional de Salud (INS), Chorrillos, Lima, Peru, 11 Ministerio de Salud Pública, Quito, Pichincha, Ecuador, 12 Laboratorio Central de Salud Pública (LCSP), Ascuncion, Distrito Capital, Paraguay, 13 Instituto Nacional de Salud (INS), Bogota, Cundinamarca, Colombia, 14 Instituto Adolfo Lutz (IAL), Sao Paulo, Sao Paulo, Brazil, 15 Division of Postgraduate Studies, Faculty of Medicine, National Autonomous University of Mexico, Mexico City, Mexico

* rpalekar@gmail.com

**Data Availability Statement:** All relevant data are within the manuscript and its Supporting Information files.

## Abstract

### Objective

Since the 2009 influenza pandemic, Latin American (LA) countries have strengthened their influenza surveillance systems. We analyzed influenza genetic sequence data from the 2017 through 2018 Southern Hemisphere (SH) influenza season from selected LA countries, to map the availability of influenza genetic sequence data from, and to describe, the 2017 through 2018 SH influenza seasons in LA.

### Methods

We analyzed influenza A/H1pdm09, A/H3, B/Victoria and B/Yamagata hemagglutinin sequences from clinical samples from 12 National Influenza Centers (NICs) in ten countries (Argentina, Brazil, Chile, Colombia, Costa Rica, Ecuador, Mexico, Paraguay, Peru and Uruguay) with a collection date from epidemiologic week (EW) 18, 2017 through EW 43, 2018. These sequences were generated by the NIC or the WHO Collaborating Center (CC) at the U.S Centers for Disease Control and Prevention, uploaded to the Global Initiative on Sharing All Influenza Data (GISAID) platform, and used for phylogenetic reconstruction.

**Funding:** The authors received no specific funding for this work.

**Competing interests:** The authors have declared that no competing interests exist.

## Findings

Influenza hemagglutinin sequences from the participating countries (A/H1pdm09 n = 326, A/H3 n = 636, B n = 433) were highly concordant with the genetic groups of the influenza vaccine-recommended viruses for influenza A/H1pdm09 and influenza B. For influenza A/H3, the concordance was variable.

## Conclusions

Considering the constant evolution of influenza viruses, high-quality surveillance data—specifically genetic sequence data, are important to allow public health decision makers to make informed decisions about prevention and control strategies, such as influenza vaccine composition. Countries that conduct influenza genetic sequencing for surveillance in LA should continue to work with the WHO CCs to produce high-quality genetic sequence data and upload those sequences to open-access databases.

## Introduction

Historically, developing countries, including those in Latin American (LA), have contributed less surveillance data than developed countries to the global understanding of patterns of influenza circulation [1,2]. Since the 2009 influenza H1N1 pandemic, however, LA countries have strengthened their influenza surveillance systems according to global and regional standards [2]. According to these published global and regional standards [3,4], countries should have active influenza surveillance systems that routinely identify persons that meet a standard case definition at sentinel sites and collect epidemiologic data and a clinical sample from these cases [3,4]. The clinical samples should be tested using influenza-sensitive and specific methods, such as real-time reverse transcriptase polymerase chain reaction (rRT-PCR), and all epidemiologic and virologic data should be analyzed on a weekly basis and be publicly disseminated [3,4].

As reflection of these advances in LA, there are currently more than 500 severe acute respiratory infection (SARI) sentinel sites conducting active influenza surveillance, 24 laboratories using molecular methods to detect influenza viruses, and more than 15 countries routinely sharing epidemiologic and virologic influenza surveillance data with the World Health Organization (WHO) on a routine basis [2].

Within the laboratory network in LA, there are currently 22 laboratories designated as National Influenza Centers (NICs) and one WHO Collaborating Center (CC) for Influenza Surveillance [U.S. Centers for Disease Control and Prevention (U.S. CDC)]; these laboratories are part of the Global Influenza Surveillance and Response System (GISRS) that includes 174 NICs and 6 WHO CCs [5]. All of these laboratories comply with pre-specified terms of reference, established by the WHO. These NICs receive samples from sentinel sites conducting influenza surveillance as well as samples from clinicians collecting samples for clinical testing and use a combination of indirect immunofluorescence and rRT-PCR to test for influenza viruses [5,6]. rRT-PCR testing is conducted using detection kits provided by the WHO CC at the U.S. CDC. Influenza A viruses are further subtyped and influenza B viruses are genotyped, by rRT-PCR [6]. The number of positive influenza samples as well as the total number of samples tested for influenza are reported on a weekly basis to the GISRS network's online platform FluNet through the Pan American Health Organization (PAHO)/WHO [7,8]. Several of the

LA NICs also use antigenic characterization methods to compare circulating influenza viruses to the influenza viruses recommended for inclusion in the influenza vaccine. In recent years, as the technology of genetic sequencing has become more widely available, approximately 25% of LA NICs have incorporated this technique into their virologic surveillance as well (personal communication, Pan American Health Organization, January 2019), given that this technique can provide detailed genetic characterization of influenza viruses.

Considering the constant evolution of influenza viruses, real-time, high-quality surveillance data, specifically genetic sequence data, are needed to allow public health decision makers to make more informed decisions about prevention and control strategies, such as influenza vaccine composition. However, as mentioned, the majority of LA NICs do not have this capacity. In order to map the availability of LA genetic sequence data and to describe the 2017 Southern Hemisphere through the 2018 Southern Hemisphere influenza seasons in LA, we analyzed the genetic sequence data available during this period from selected LA countries.

## Methods

### Country selection

PAHO is the WHO Regional Office for the Americas and provides technical cooperation to countries in the Americas to strengthen their influenza surveillance systems. Of the 22 NICs in LA, PAHO invited 12 NICs in ten countries (Argentina-Buenos Aires, Brazil-Pará, Brazil-Rio de Janeiro, Brazil-Sao Paolo, Chile, Colombia, Costa Rica, Ecuador, Mexico, Paraguay, Peru and Uruguay) to participate in a bioinformatics training course conducted by the WHO CC at the U.S. CDC at PAHO. These NICs were subsequently invited to participate in an analysis describing the genetic evolution of influenza viruses during the 2017 through 2018 Southern Hemisphere influenza seasons (Table 1).

### Period of analysis

Clinical samples with a date of collection of May 1, 2017 through October 26, 2018, representing epidemiologic week (EW) 18, 2017 to EW 43, 2018, that were uploaded to the Global Initiative on Sharing All Influenza Data (GISAID) platform by October 28, 2018, were included in the analysis, based upon the fact that the South Hemisphere influenza season typically peaks in EW 18 and ends in EW 36 [8–10].

### Virologic data

Virologic data for the period of analysis were downloaded for the participating countries from the WHO open-access database FluNet [8]. This dataset included the number of samples tested for influenza and the number of samples positive for influenza by type/subtype, by EW of symptom onset.

### Genetic sequence data

GISAID maintains a publicly accessible database of uploaded influenza genetic sequences and is the most commonly used public database for the storage of genetic sequences among laboratories participating in GISRS [11].

As part of the NIC terms of reference, NICs routinely share clinical samples with the WHO CCs for additional antigenic and genetic characterization [7]. The WHO CC at the U.S. CDC routinely completes whole genome genetic sequencing from the samples received from the NIC's in LA and uploads the sequences to GISAID. NICs that do *in-situ* genetic sequencing of

**Table 1. Sequencing capacity and number of hemagglutinin (HA) genetic sequences available in GISAID—participating National Influenza Centres, May 1, 2017 to October 26, 2018.**

| Country | Institution | Sanger sequencing capacity | Currently sequencing viruses | Total number of sequences available in GISAID | Total number of sequences uploaded to GISAID by the NIC[a] | Total number of sequences included in the study[b] |
|---|---|---|---|---|---|---|
| Argentina | Instituto Nacional de Enfermedades Infecciosas, ANLIS C.G. Malbran | Y | Y | 240 | 40 | 88 |
| Brazil | Fundação Oswaldo Cruz | Y | Y | 348 | 240 | 291 |
| Brazil | Instituto Adolfo Lutz | Y | Y | 247 | 102 | 190 |
| Brazil | Instituto Evandro Chagas | Y | Y | 115 | 62 | 89 |
| Chile | Instituto de Salud Publica de Chile | Y | Y | 390 | 223 | 321 |
| Colombia | Instituto Nacional de Salud | Y | Y[c] | 92 | 0 | 78 |
| Costa Rica | Instituto Costarricense de Investigación y Enseñanza en Nutrición y Salud | N | Y[c] | 36 | 0 | 33 |
| Ecuador | Instituto Nacional de Investigación en Salud Pública | N | Y[c] | 61 | 0 | 61 |
| Mexico | Instituto de Diagnostico y Referencia Epidemiologicos | Y | Y | 170 | 62 | 45 |
| Paraguay | Laboratorio Central de Salud Pública | N | Y[c] | 62 | 19 | 60 |
| Peru | Instituto Nacional de Salud | Y | Y[c] | 107 | 0 | 87 |
| Uruguay | Departamento de Laboratorio de Salud Publica | N | Y[c] | 62 | 13 | 52 |
| TOTAL | - | - | - | 1930 | 761 | 1395 |

[a] Sequences uploaded from May 1 2017 to October 28 2018

[b] Sequences with optimal length and without gaps and mismatches obtained from original samples

[c] Country uses external sequencing service

influenza viruses also routinely upload their sequences to GISAID and were asked to do so, by October 28, 2018.

All influenza A and B hemagglutinin (HA) sequences available in GISAID from the participating countries were downloaded for the period of analysis on October 28, 2018 and their accession numbers were recorded (S1 Table). These sequences were uploaded either by the WHO CC at the U.S. CDC or by the participating NIC, depending on which institution completed the genetic sequencing. Sequences available in GISAD from non-participating countries in LA and the Caribbean for the study period were also downloaded and included in the HA alignment datasets.

## Alignment of sequences

The HA sequences of influenza A/H1pdm09, A/H3, B/Victoria and B/Yamagata cases were aligned using the Clustal W integrated tool within BioEdit [12], along with their respective reference strains, provided by the WHO CC at the U.S. CDC. Only sequences obtained from original clinical samples were included in the analysis; sequences obtained from multiple virus-passages, incomplete HA sequences, or HA sequences with mismatches/gaps were excluded.

The curated HA alignment datasets of each virus were submitted, in order to estimate the maximum likelihood phylogenetic trees, to the TREESUB phylogenetic program using RAxML and PAML, followed by branch annotation of amino acid substitutions. The general

time reversible+Γ (GTR+GAMMA) nucleotide substitution model was selected in RAxML v.7.3.0 for tree inference. Ancestral codon substitutions for each gene were estimated using baseml, as implemented in PAML8 using the ML trees inferred. Non-synonymous substitutions were then transcribed onto the consensus gene phylogenies and visualized in FigTree v1.4.3 [13]. Clusters were defined as a clade in the tree with discrete amino acid differences when compared to the root sequence of the tree. The frequency of influenza virus genetic groups was analyzed using Tableau software [14].

## Influenza vaccine composition

Each country provided information about the influenza vaccine (Northern versus Southern Hemisphere composition and trivalent versus quadrivalent composition) that was used during the study period.

## Results

### Genetic sequence data set

Seven hundred and sixty-one influenza HA sequences were produced and uploaded by the participating NICs to the GISAID database—Argentina (n = 40), Brazil NIC-FIOCRUZ (n = 240), Brazil NIC-IAL (n = 102) and Brazil NIC-IEC (n = 62), Chile (n = 223), Mexico (n = 62), Paraguay (n = 19) and Uruguay (n = 13). An additional n = 1,169 sequences were uploaded by the WHO CC at the U.S. CDC to GISAID from the samples collected by the participating countries (Table 1).

After application of the exclusion criteria, a total of n = 1,395 HA sequences were included in the final phylogenetic analysis of influenza A/H1pdm09, influenza A/H3, influenza B/Yamagata, and influenza B/Victoria viruses (Table 1). This total included n = 836 sequences completed by the WHO CC at the U.S. CDC and n = 559 sequences completed by the participating countries.

### Patterns of influenza virus circulation

During the period of analysis, influenza A viruses predominated over influenza B. Among the subtyped influenza A viruses, during the 2017 Southern Hemisphere and the 2017–18 Northern Hemisphere seasons, influenza A/H3 predominated. During the 2018 Southern Hemisphere season, influenza A/H1pdm09 predominated. Among the influenza B viruses with lineage information, during all three seasons, B/Yamagata predominated (Fig 1).

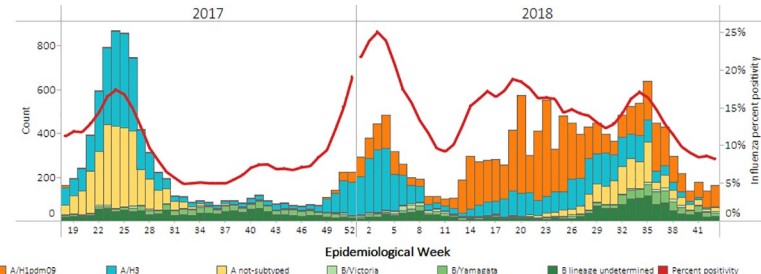

**Fig 1. Patterns of influenza circulation among selected countries in Latin America [a], epidemiologic weeks 18, 2017 through 43, 2018.** [a]Argentina, Brazil, Chile, Colombia, Costa Rica, Ecuador, Mexico, Paraguay, Peru, and Uruguay.

## Influenza A(H1N1)pdm09 HA genetic analysis

A total of n = 326 HA sequences from participating countries were included in the phylogenetic analysis of influenza A/H1N1pdm09. Phylogenetic analysis showed that these sequences grouped within HA subclade 6B.1, characterized by the S84N, S162N, I216T substitutions (Fig 2). Within this subclade, the majority of sequences (n = 320) were clustered in the subclade 6B.1A that share the S164T substitution. Among these sequences, most (n = 82) were from the

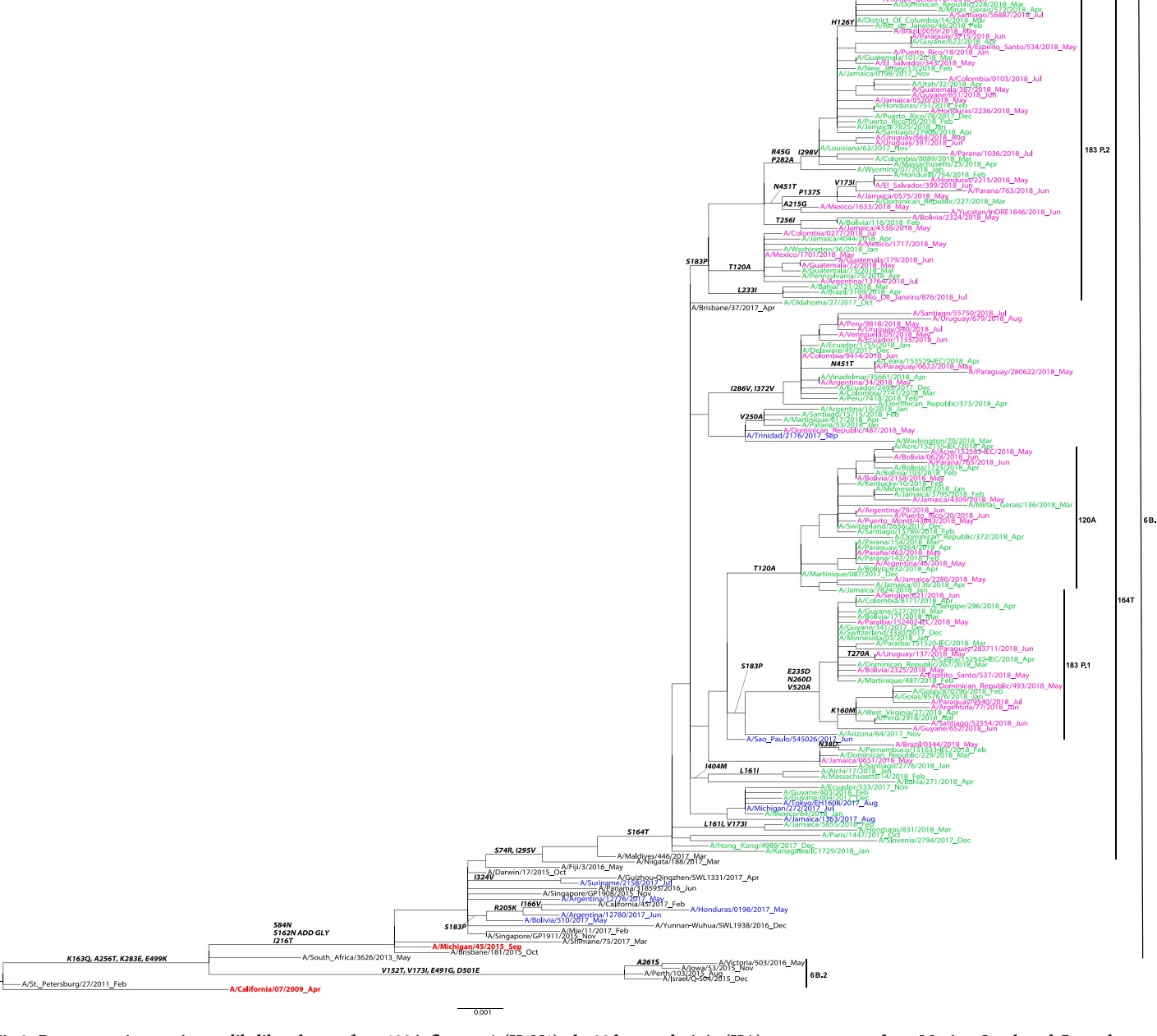

**Fig 2. Representative maximum-likelihood tree of n = 139 influenza A (H1N1)pdm09 hemagglutinin (HA) gene sequences from Mexico, South and Central America; sequences from the current and previous vaccine strains (in red) and reference viruses detected worldwide indicated by the CDC WHO CC.** HA sequences of influenza viruses collected from May to September 2017 are in blue, October 2017 to April 2018 are in green, May to September 2018 are in pink. Sequences from the time period before the period of analysis, are in black. Amino acid changes and addition (ADD GLY) and loss (LOSS GLY) of glycosylation sites are indicated in bold in the branches.

subclade 183P-1, followed by subclade 6B.1A/183P-2 (n = 58), and then 6B.1A/183P-3 (n = 50) (Table 2).

## Influenza A(H3N2) HA genetic analysis

A total of n = 636 HA sequences from participating countries were included in the phylogenetic analysis of influenza A/H3 (Fig 3). Phylogenetic analysis showed that the sequences belonged to the HA genetic groups 3C.2a (n = 586) and 3C.3a (n = 50). Among the 3C.2a clade viruses, most of the sequences (n = 530), clustered in the subclades 3C.2a2 (n = 352) and 3C.2a1, and its subclades (n = 178) (Table 2). Within the genetic group 3C.2a1, the majority of these sequences belonged to the 3C.2a1b subclade defined by the amino acid substitutions N171K, I406V, G484E. All sequences in the 3C.2a1b/135K subclade were collected during 2018 and grouped in a smaller subclade sharing an amino acid substitution T128A on HA1 leading to a loss of a glycosylation motif. Of note, none of the sequences from the participating countries clustered in the 3C.2a1b/135N subclade.

## Influenza B virus HA genetic analysis

A total of n = 433 HA sequences from the participating countries were included in the phylogenetic analysis of influenza B—n = 350 Yamagata-lineage sequences and n = 83 Victoria-lineage sequences. All influenza B/Yamagata HA sequences grouped in the Y3 clade sharing the S150I, N165Y and G229D substitutions as compared to the B/Florida/4/2006 vaccine strain (Fig 4). All influenza B/Victoria HA sequences belonged to clade V1A with HA1 substitutions

**Table 2. Hemagglutinin amino acid substitutions compared to reference influenza virus vaccine strain—Latin America and the Caribbean, May 1, 2017 to October 26, 2018.**

| Influenza virus | Reference vaccine virus | Genetic group | Signature amino acid substitution[a] | Antigenic site | Collection date range[b] | Season | Geographic location (Number of sequences) | | |
|---|---|---|---|---|---|---|---|---|---|
| | | | | | | | Participating countries | Other countries from Americas[c] | Total number of sequences |
| **H1pdm09** | A/ Michigan/ 45/2015 | 6B.1 | S84N **S162N** I216T | - Sa - | May 2017 to Jul 2017 | SH 2017 | Argentina, Brazil (n = 5) | Bolivia, Honduras, Suriname (n = 14) | 19 |
| | | 6B.1A | S164T[c] | Sa | Jun 2017 to Aug 2018 | SH 2017; NH 2017– 2018; SH 2018 | Argentina, Brazil, Chile, Colombia, Ecuador, Mexico, Paraguay, Peru, Uruguay (n = 130) | Dominican Republic; French Guiana; Honduras; Jamaica; Martinique; Trinidad and Tobago; Venezuela (n = 25) | 155 |
| | | 6B.1A/ 183P-1 | S183P | Sa | Dec 2017 to Jul 2018 | NH 2017– 2018; SH 2018 | Argentina, Brazil, Chile, Colombia, Paraguay, Peru, Uruguay (n = 82) | Bolivia, Dominican Republic, French Guiana, Martinique (n = 15) | 97 |
| | | 6B.1A/ 183P-2 | S183P L233I | Sa | Nov 2017 to Aug 2018 | NH 2017– 2018; SH 2018 | Brazil, Chile, Colombia, Mexico, Paraguay, Uruguay (n = 58) | Bolivia, Dominican Republic; El Salvador, French Guiana; Honduras; Jamaica; Puerto Rico (n = 126) | 184 |
| | | 6B.1A/ 183P-3 | S183P T120A | | Dec 2017 to Aug 2018 | NH 2017– 2018; SH 2018 | Argentina, Brazil, Chile, Colombia, Mexico, Paraguay (n = 50) | Bolivia, Dominican Republic, Guatemala, Jamaica, Martinique, Puerto Rico (n = 55) | 105 |

*(Continued)*

**Table 2.** (*Continued*)

| Influenza virus | Reference vaccine virus | Genetic group | Signature amino acid substitution[a] | Antigenic site | Collection date range[b] | Season | Geographic location (Number of sequences) | | Total number of sequences |
|---|---|---|---|---|---|---|---|---|---|
| | | | | | | | Participating countries | Other countries from Americas[c] | |
| **H3** | A/Texas/50/2012 | 3C.2a | L3I<br>N144S<br>S159Y<br>V186G<br>Q311H<br>D489N | -<br>A<br>B<br>B<br>-<br>- | Jun 2017 | SH 2017 | - | Bolivia (n = 1) | 1 |
| | | 3C.2a1 | N171K, I406V, G484E | -<br>-<br>- | May to Aug 2017 | SH 2017 | Brazil, Chile, Colombia, Mexico, Peru, (n = 49) | Panama (n = 1) | 50 |
| | | 3C.2a1a | G479E | - | May 2017 to Jan 2018 | SH 2017; NH 2017–2018 | Argentina, Brazil, Chile, Colombia, Costa Rica, Mexico, Peru, Uruguay (n = 45) | Bolivia, Panama; Puerto Rico (n = 13) | 58 |
| | | 3C.2a1b | K92R<br>H311Q | E<br>- | May to Jul 2018 | SH 2018 | Brazil, Chile, Colombia, Costa Rica, Ecuador, Mexico, Paraguay, Peru (n = 64 | Bolivia, Dominican Republic; El Salvador, Guadeloupe, Guatemala. Haiti, Honduras, Jamaica; Mexico, Nicaragua, Puerto Rico (n = 93) | 157 |
| | | 3C.2a1b/135K | **T135K** | A | Jun 2017 to Jul 2018 | SH 2017, NH 2017–2018, SH 2018 | Brazil, Chile (n = 20) | Bolivia, Puerto Rico (n = 6) | 26 |
| | | 3C.2a2 | T131K<br>R142K<br>R261Q | A<br>A<br>- | May 2017 to Aug 2018 | SH 2017, NH 2017–2018, SH 2018 | Argentina, Brazil, Chile, Colombia, Costa Rica, Ecuador, Mexico, Paraguay, Peru, Uruguay (n = 352) | Bolivia, Dominican Republic; French Guiana, Guatemala. Haiti, Honduras, Jamaica; Martinique, Nicaragua, Panama, Puerto Rico, Suriname (n = 159) | 511 |
| | | 3C.2a3 | N121K<br>S144K | D<br>A | May 2017 to Apr 2018 | SH 2017, NH 2017–2018 | Brazil, Chile, Mexico (n = 53) | Bolivia, Puerto Rico (n = 6) | 59 |
| | | 3C.2a4 | N31S<br>D53N<br>R142G<br>S144R<br>K160T<br>N171K<br>I192K<br>Q197H | -<br>C<br>A<br>A<br>B<br>-<br>B<br>B | May to Jul 2017 | SH 2017 | Brazil, (n = 3) | Honduras (n = 3) | 6 |
| | | 3C.3a | **T128A**<br>A138S<br>R142G | A<br>A<br>A | May 2017 to Aug 2018 | SH 2017, NH 2017–2018, SH 2018 | Brazil, Colombia, Paraguay, Peru, Uruguay (n = 50) | Guatemala, Jamaica, Puerto Rico (n = 11) | 61 |

(*Continued*)

**Table 2.** (Continued)

| Influenza virus | Reference vaccine virus | Genetic group | Signature amino acid substitution[a] | Antigenic site | Collection date range[b] | Season | Geographic location (Number of sequences) | | Total number of sequences |
|---|---|---|---|---|---|---|---|---|---|
| | | | | | | | Participating countries | Other countries from Americas[c] | |
| B–VIC | B/Colorado/06/2017 | 1A | | | May 2017 to May 2018 | SH 2017, NH 2017–2018, SH 2018 | Brazil, Chile, Costa Rica, Mexico, Paraguay, Peru, Uruguay (n = 26) | Bolivia, Dominican Republic, Guadeloupe. Panama, Puerto Rico (n = 19) | 45 |
| | | 1A.1 | I180V N162Δ N163Δ R498K | - 160 loop 160 loop - | May 2017 to May 2018 | SH 2017, NH 2017–2018, SH 2018 | Argentina, Brazil, Chile, Costa Rica, Mexico, Paraguay, Peru, Puerto Rico (n = 57) | Barbados, Bolivia, Dominican Republic, El Salvador, French Guiana, Guatemala, Haiti, Honduras, Jamaica, Martinique, Panama, Puerto Rico, Suriname, Trinidad and Tobago (n = 121) | 178 |
| B–YAM | B/Florida/4/2006 | Y3 | S150I N163Y G229D | 150 loop 160 loop - | May 2017 to Aug 2018 | SH 2017, NH 2017–2018, SH 2018 | Argentina, Brazil, Chile, Colombia, Costa Rica, Ecuador, Mexico, Paraguay, Peru, Uruguay (n = 350) | Barbados, Bolivia, Dominican Republic; El Salvador, French Guiana, Guatemala. Haiti, Honduras, Jamaica; Martinique, Nicaragua, Panama, Puerto Rico (n = 257) | 607 |

[a] Bold text indicates loss of glycosylation site.

[b] Collection date range for all sequences.

[c] Close identity sequences

Δ amino acid deletion

I117V, N129D and V146I compared to vaccine virus, B/Brisbane/60/2008 (Fig 5). Among the B/Victoria V1A clade, one major subclade with the two amino acid deletions at positions K162 and N163 of HA1, that defines the V1A.1 genetic group, was identified. Of note, none of the sequences from the participating countries clustered in the V1A-3 DEL subcluster that has the K162, N163 and D164 triple deletion.

### Influenza virus genetic characterization among participating countries and other Latin American and Caribbean countries

All influenza A/H1pdm09 viruses were from the 6B.1 genetic group and over the period of analysis, the frequency of the 6B.1A clade increased (Fig 6A). Additionally, over the period of analysis, there was diversification of the 6B.1A subclade, with a higher frequency of the sub-clade 6B.1A/183P-2 starting at the end of 2017 and throughout 2018. Globally, similar increases in circulating viruses in the 6B.1A/183P subclades were observed (Fig 7) [15].

Among the influenza A/H3 viruses, within the 3C.2a genetic group, the overall predominant subclade, was 3C.2a2, but the frequency of the subclade 3C.2a1b/135K increased during the period of analysis (Fig 6B). The frequency of the 3C.3a subclade increased at the end of the 2018 period of analysis. Although viruses belonging to the subclade 3C.2a1b/135N circulated globally, the circulation of subclade 3C.2a1b/135K also increased globally, similar to the pattern observed in LA (Fig 7).

Among influenza B viruses, influenza B/Victoria genotype V1A.1 with a double amino acid deletion (162/163), increased over the study period, replacing the V1A genetic group (Fig 6C).

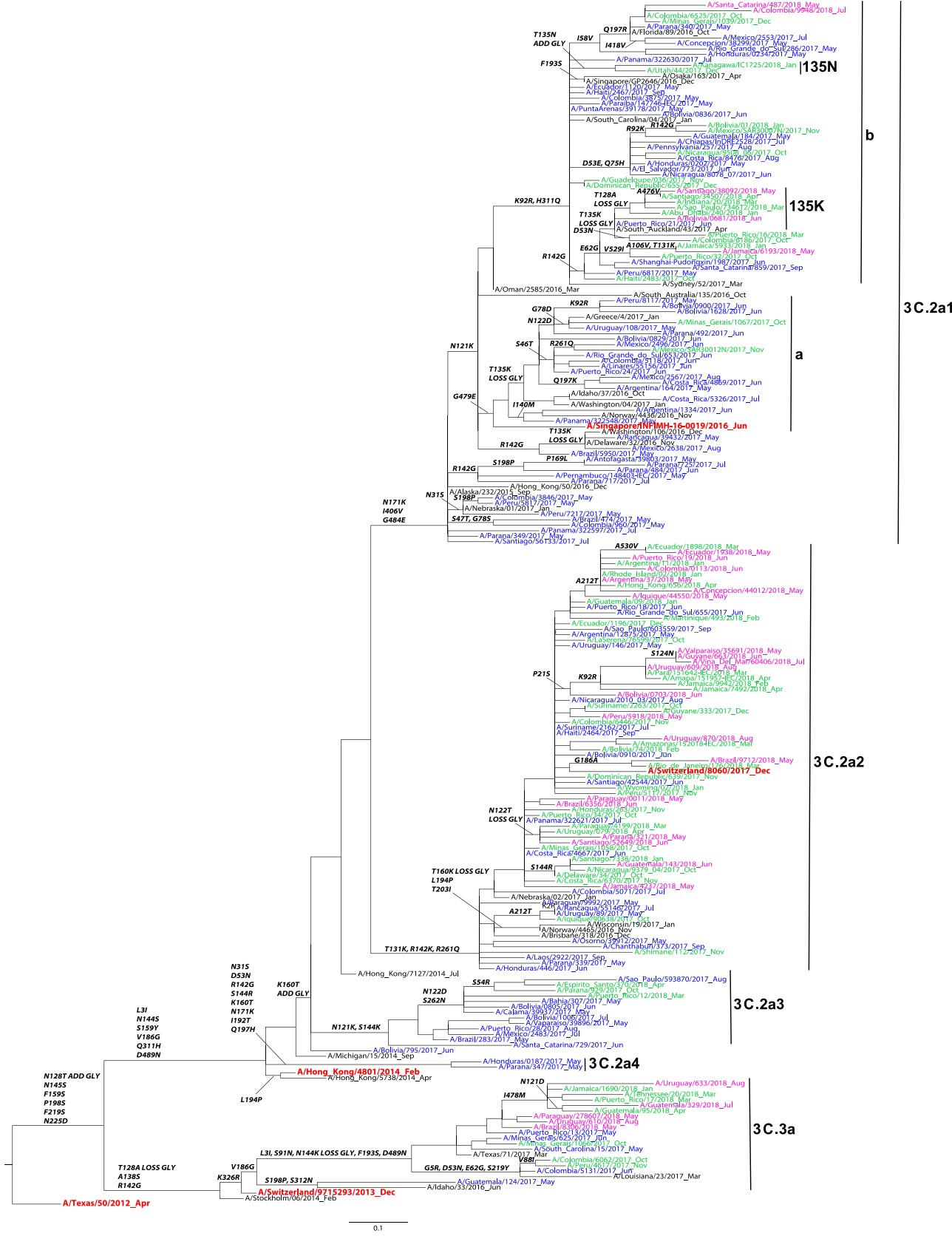

**Fig 3. Representative maximum-likelihood tree of n = 180 influenza A (H3N2) HA gene sequences from Mexico, South and Central America; sequences from the current and previous vaccine strains (in red) and reference viruses detected worldwide indicated by the CDC WHO CC.** HA sequences of influenza viruses collected from May to September 2017 are in blue, October 2017 to April 2018 are in green, May to September 2018 are in pink. Sequences from the time period before the period of analysis, are in black. Amino acid changes and addition (ADD GLY) and loss (LOSS GLY) of glycosylation sites are indicated in bold in the branches.

Globally, similar replacement of the influenza B/Victoria V1A with no amino acid deletion with V1A.1 with a double amino acid deletion was observed (Fig 7). Influenza B/Yamagata viruses did not show much genetic evolution over the period of analysis, and the Y3 genetic group predominated in the analysis as well as globally (Fig 6D and Fig 7).

**Influenza vaccine composition.** Argentina, Brazil, Chile, Colombia, Costa Rica, Paraguay, Peru and Uruguay used the Southern Hemisphere trivalent vaccine during the 2017 and 2018 Southern Hemisphere influenza seasons and Ecuador and Mexico used the Northern Hemisphere trivalent vaccine during the 2017–18 Northern Hemisphere influenza season.

**Comparison of predominant genetic groups to vaccine-recommended virus genetic groups:** Among the influenza A/H1pdm09 viruses, the 6B.1 subclade predominated during all three influenza seasons and was the subclade recommended for inclusion in the influenza vaccine during all three influenza seasons (Table 3). Among the influenza A/H3 viruses, the 3C.2a2 subclade predominated during all three influenza seasons; the vaccine-recommended virus for the first two seasons was a 3C.2a virus and in the last season was a 3C.2a1 virus (Table 3). Among the influenza B/Victoria viruses, the clade that predominated during the 2017 Southern Hemisphere season was V1A, which was the clade of the vaccine-recommended virus; the subclade that predominated during the last two seasons was V1A.1, while the vaccine-recommended virus continued to be from the V1 clade (Table 3). Among the influenza B/Yamagata viruses, the clade that predominated over the period of analysis was Y3, which was the vaccine-recommended virus for all three seasons. Of note, while B/Yamagata viruses predominated over B/Victoria viruses during all three seasons of the analysis, the trivalent vaccine used in the participating countries, during the first two seasons of the analysis (2017 Southern Hemisphere and 2017–18 Northern Hemisphere) only contained a B/Victoria virus (Table 3).

## Discussion

This is the first published analysis describing the patterns of influenza circulation in LA using genetic sequence data. There are three key findings from this analysis. First, our analysis showed that the viruses that circulated in these countries during the early part of the 2017 Southern Hemisphere influenza season evolved and changed as compared to those that circulated at the end of the 2018 Southern Hemisphere season, likely due to antigenic drift. Second, the viruses that circulated in these countries during the 2017 through the 2018 Southern Hemisphere season, while varying in predominance, resembled those detected globally [16–19]. Finally, the genetic groups that predominated in this analysis matched, to varying extents, the genetic groups of the influenza vaccine-recommended viruses for the 2017 through the 2018 Southern Hemisphere seasons [16–19].

With regard to the concurrence between the A/H1pdm09 influenza vaccine-recommended viruses and the viruses that predominated in this analysis, during all three seasons analyzed, the vaccine-recommended virus was A/Michigan/45/2015(H1N1)pdm09-like virus, which belongs to the 6B.1 subclade, which was the subclade that predominated in our analysis. While we do not present antigenic characterization of H1pdm09 viruses from these countries in this analysis, other published analyses have documented the inhibition of 6B.1 viruses with ferret anti-sera raised against A/Michigan/45/2015(H1N1)pdm09-like virus [16–19].

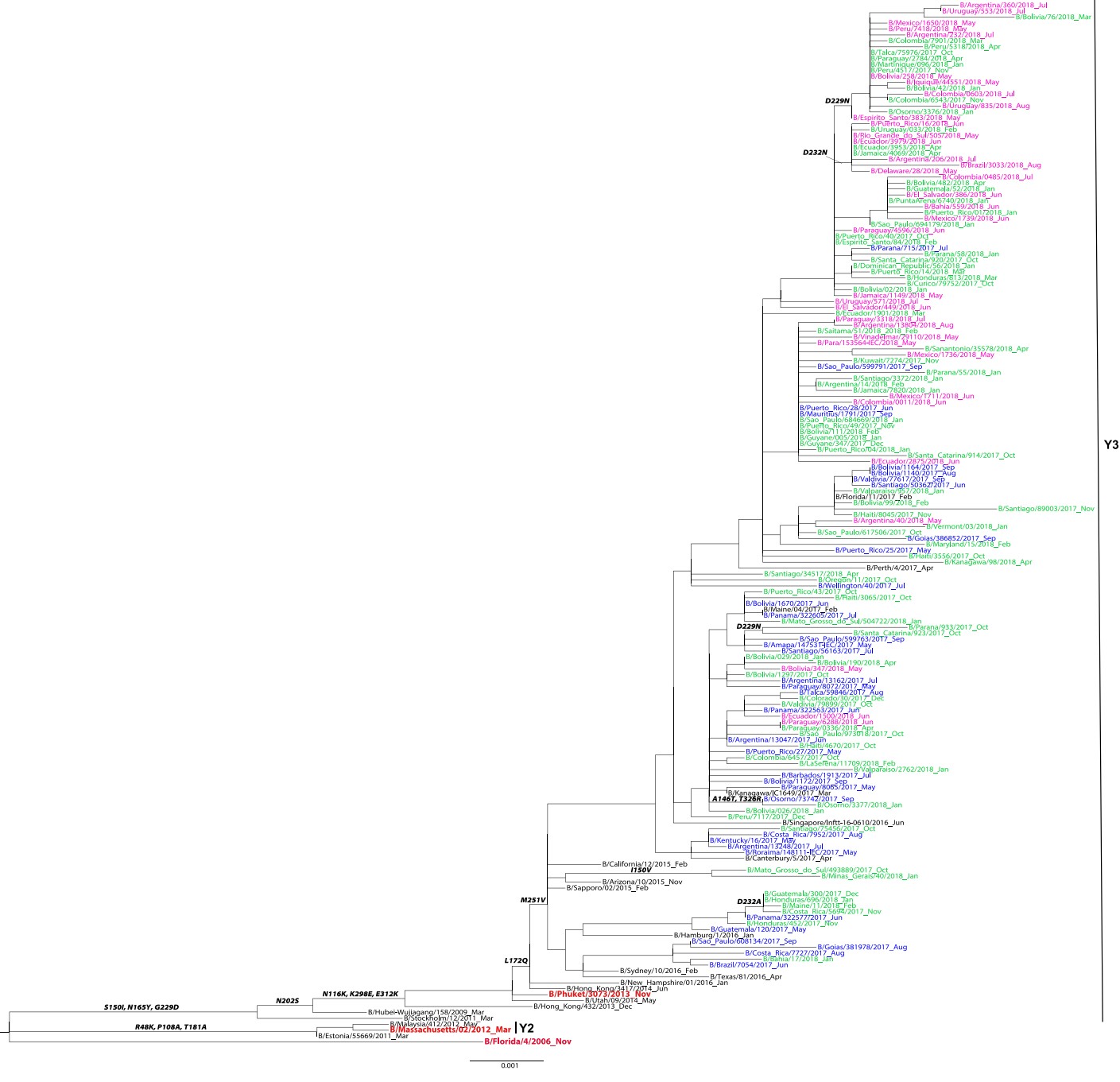

**Fig 4. Representative maximum-likelihood tree n = 141 influenza B virus Yamagata HA gene sequences from in Mexico, South and Central America; sequences from the current and previous vaccine strains (in red) and reference viruses detected worldwide indicated by the CDC WHO CC.** HA sequences of influenza viruses collected from May to September 2017 are in blue, October 2017 to April 2018 are in green, May to September 2018 are in pink. Sequences from the time period before the period of analysis, are in black. Amino acid changes and addition (ADD GLY) and loss (LOSS GLY) of glycosylation sites are indicated in bold in the branches.

With regard to the concurrence between the A/H3 influenza vaccine- recommended viruses and the viruses that predominated in this analysis, during the 2017 Southern Hemisphere and 2017–18 Northern Hemisphere influenza seasons, the vaccine recommended virus

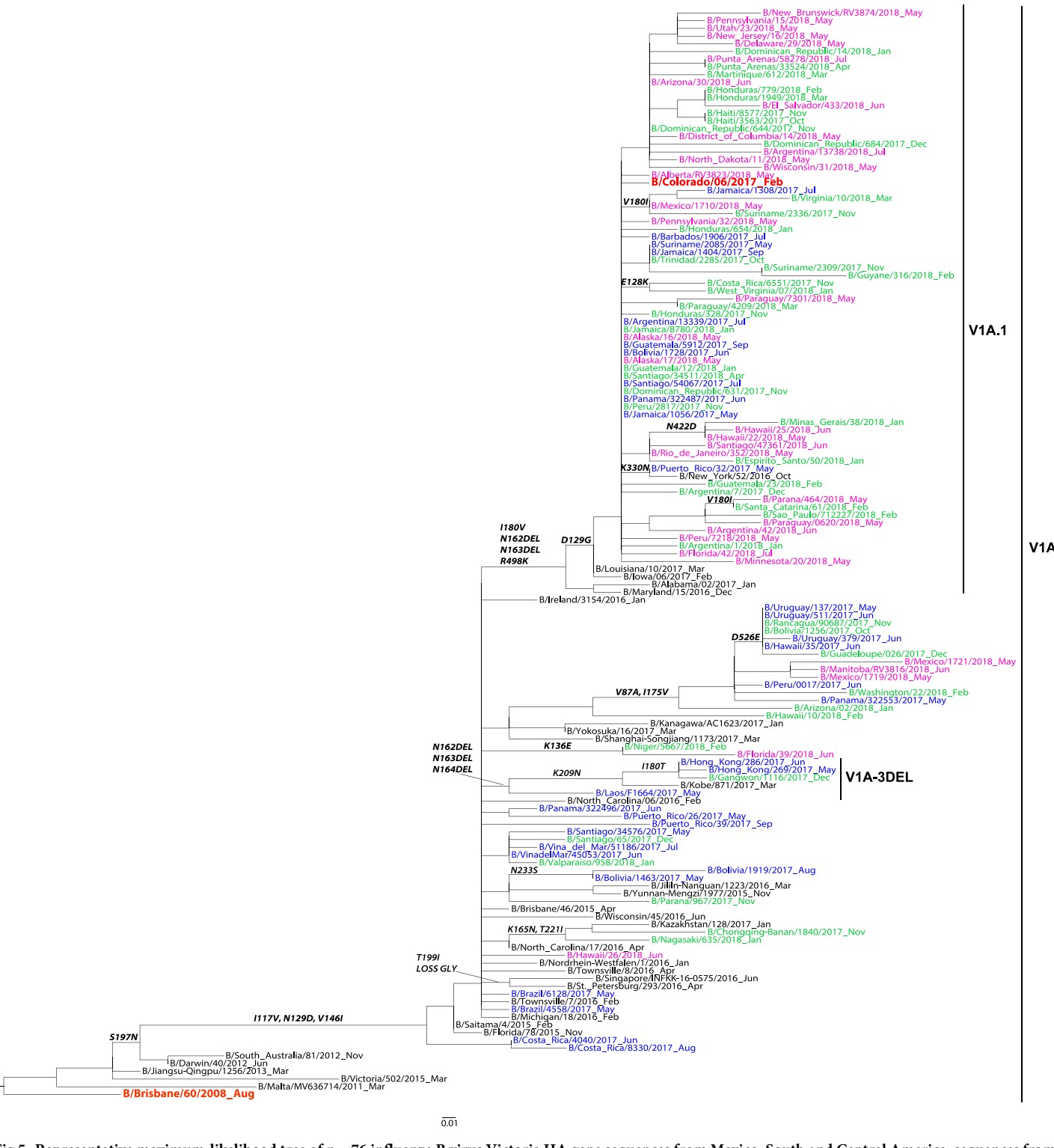

**Fig 5. Representative maximum-likelihood tree of n = 76 influenza B virus Victoria HA gene sequences from Mexico, South and Central America; sequences from the current and previous vaccine strains (in red) and reference viruses detected worldwide indicated by the CDC WHO CC. HA sequences of influenza viruses collected from May to September 2017 are in blue, October 2017 to April 2018 are in green, May to September 2018 are in pink. Sequences from the time period before the period of analysis, are in black. Amino acid changes and addition (ADD GLY) and loss (LOSS GLY) of glycosylation sites are indicated in bold in the branches**.

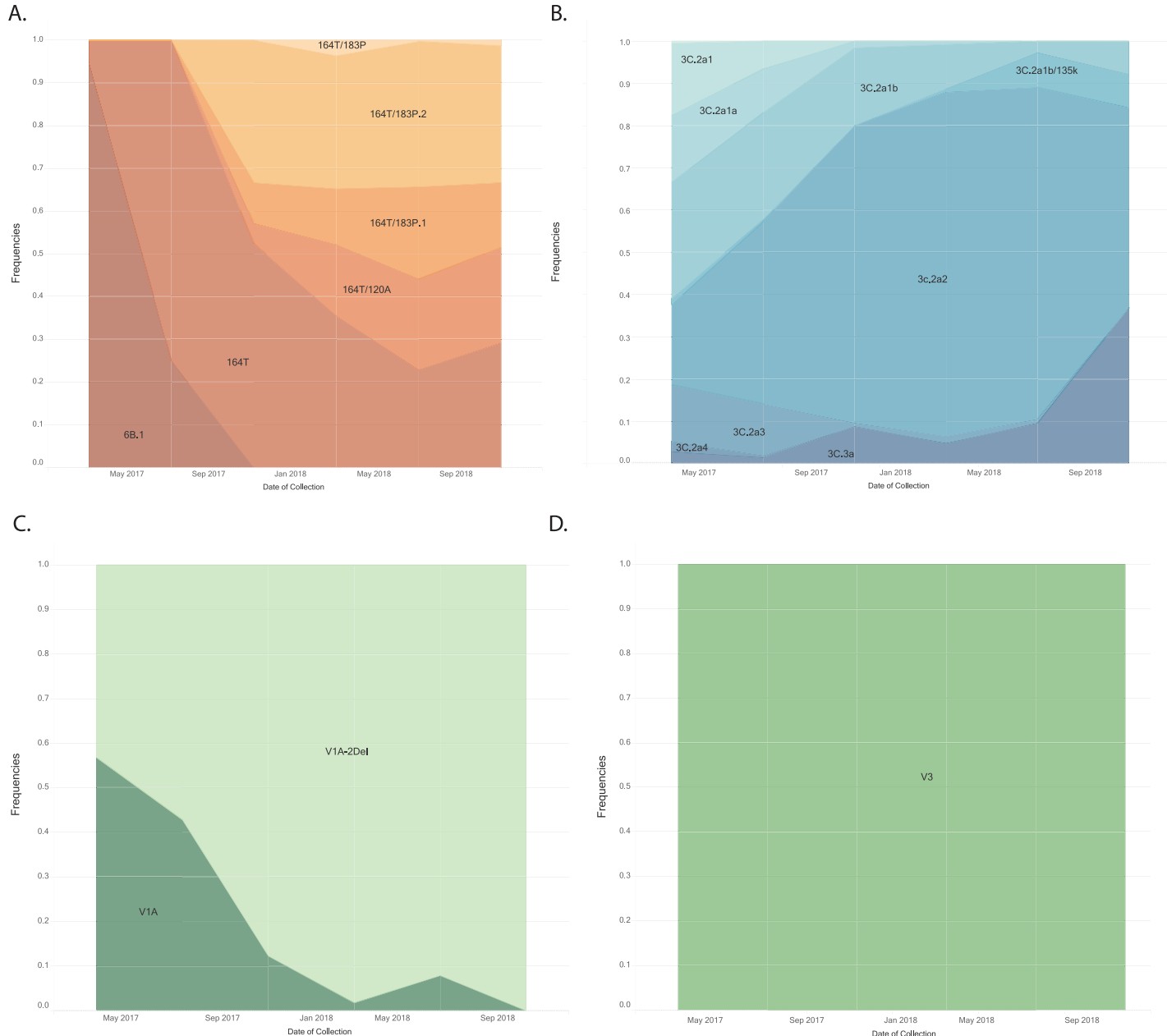

**Fig 6.** Frequency of genetic groups of influenza viruses A/H1pdm09 (A), A/H3 (B), B/Victoria (C) and B/Yamagata (D) based upon hemagglutinin (HA) gene sequences from participating countries and other Latin American and Caribbean countries, May 1, 2017 through October 26, 2018.

was A/Hong Kong/4801/2014(H3N2)-like virus, a 3C.2a clade virus. During this period in our analysis, the subclade that predominated was 3C.2a2; and while we do not present antigenic characterization of H3 viruses from these countries in this analysis, other published analyses have documented lower inhibition of egg-propagated 3C.2a2 viruses with ferret anti-sera raised against A/Hong Kong/4801/2014(H3N2)-like virus [17–19]. During the 2018 Southern Hemisphere season, the vaccine recommended virus was A/Singapore/INFIMH-16-0019/2016(H3N2)-like virus, a 3C.2a1 subclade virus. During this period in our analysis, the subclade that predominated continued to be 3C.2a2; and while we do not present antigenic

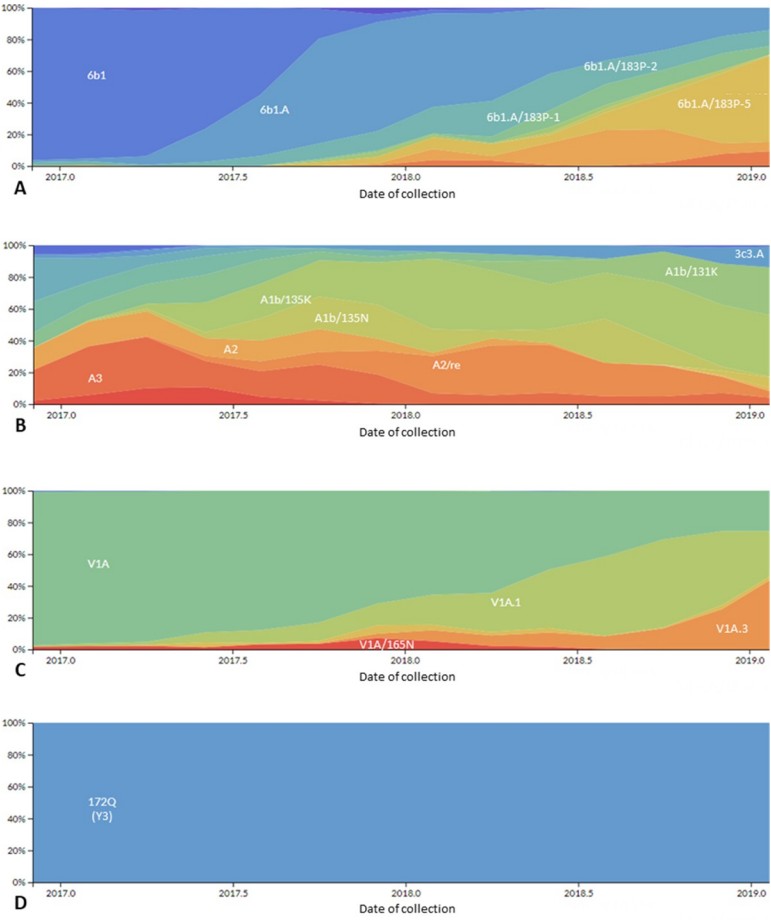

**Fig 7.** Frequency of genetic groups of globally circulating influenza viruses A/H1pdm09 (A), A/H3 (B), B/Victoria (C) and B/Yamagata (D) based upon hemagglutinin (HA) gene sequences from May 1, 2017 through October 26, 2018 obtained through Next Strain (available at nexstrain.org/flu/seasonal), accessed September 19, 2019.

characterization of H3 viruses from these countries in this analysis, other published analyses have documented a high inhibition of 3C.2a2 viruses by ferret sera raised against A/Singapore/INFIMH-16-0019/2016(H3N2)-like virus [16].

**Table 3. Predominant genetic groups of circulating viruses compared to influenza vaccine-recommended viruses.**

| Influenza virus | 2017 Southern Hemisphere influenza season | | 2017–18 Northern Hemisphere influenza season | | 2018 Southern Hemisphere influenza season | |
|---|---|---|---|---|---|---|
| | Genetic group of vaccine-recommended virus | Genetic group that predominated in analysis[b] (May-Sep 2017) | Genetic group of vaccine-recommended virus | Genetic group that predominated in analysis[b] (Oct 2017—Apr 2018) | Genetic group of vaccine-recommended virus | Genetic group that predominated in analysis[b] (May-Sep 2018) |
| H1pdm09 | 6B.1 | 6B.1 | 6B.1 | 6B.1 | 6B.1 | 6.B1. |
| H3 | 3C.2a | 3C.2a2 | 3C.2a | 3C.2a2 | 3C.2a1 | 3C.2a2 |
| B Victoria[a] | V1A | V1A | V1A | V1A.1 | V1A | V1A.1 |
| B Yamagata[a] | Y3 | Y3 | Y3 | Y3 | Y3 | Y3 |

[a] The lineage of the influenza B virus included in the 2017 Southern Hemisphere and 2017–18 Northern Hemisphere trivalent vaccine was B/Victoria, and the lineage of the influenza B virus included in the 2018 Southern Hemisphere trivalent vaccine was B/Yamagata

[b] Among sequences from participating countries (Argentina, Brazil, Chile, Colombia, Costa Rica, Ecuador, Mexico, Paraguay, Peru, Uruguay)

With regard to influenza B viruses, influenza B/Yamagata viruses predominated during the period of analysis. During all three seasons analyzed, the vaccine-recommended virus was B/Phuket/3073/2013-like virus, which belongs to the Y3 clade—the subclade that predominated in our analysis. While we do not present antigenic characterization of B Yamagata viruses from these countries in this analysis, other published analyses have documented the inhibition of Y3 viruses with ferret anti-sera raised against B/Phuket/3073/2013-like virus [16–19]. While the match between the circulating virus and the vaccine virus was good, it should be noted that none of the countries participating in this analysis used the influenza vaccine that contained this B/Yamagata virus during the first two seasons (2017 Southern Hemisphere and 2017–18 Northern Hemisphere), but rather used the trivalent vaccine that contained a B/Victoria virus. To date, observational studies have shown some cross-protection between lineages in seasons with influenza B lineage mismatch, but more analyses are needed of cross-protection as well as the cost effectiveness of the use of a quadrivalent versus a trivalent vaccine [20–23].

With regard to the concurrence between the B/Victoria influenza vaccine- recommended viruses and the viruses that predominated in this analysis, during all three seasons analyzed, the vaccine-recommended virus was B/Brisbane/60/2008-like virus, which belongs to the V1A clade, which was the subclade that predominated in our analysis only during the 2017 Southern Hemisphere influenza season. During the other two seasons (2017–18 Northern Hemisphere and 2018 Southern Hemisphere), the subclade V1A.1 predominated. While we do not present antigenic characterization of B Victoria viruses from these countries in this analysis, other published analyses have documented the inhibition of V1A viruses with ferret anti-sera raised against B/Brisbane/60/2008-like virus but limited inhibition of V1A.1 viruses [16–19].

Overall, the limited diversification of the genetic groups of influenza A/H1pdm09 and B viruses circulating in the LA during the period of analysis ressembes the slower rates of antigenic changes and evolution diversification observed globally; while the diversification we observed in LA related to H3 viruses is similar to what was observed globally [24–26].

There are two key limitations to this analysis. First, the genetic sequence data from the 12 participating NICs in LA were not necessarily from samples that were randomly selected for genetic sequencing and as such might not be representative of the influenza viruses circulating in the participating countries nor in LA overall. Second, the samples were collected during a limited period of time, intending to cover two SH influenza seasons. However, sequencing of samples collected outside of this period, could have provide additional information about the genetic evolution of the circulating influenza viruses.

In conclusion, this is the first published analysis using genetic sequence surveillance data from 10 LA countries during the 2017 through the 2018 Southern Hemisphere influenza seasons, and while there are limitations to this analysis, the public health importance of this type of analysis outweighs this, considering the prior paucity of data from LA. Increasing genetic sequencing capacity in LA is important, and standard sequencing platforms, laboratory quality assurance, use of validated protocols, sequencing and bioinformatics trainings, and support for reagent and supplies are some key components that will lead to improvements. This capacity for genetic sequence surveillance is new in LA, and countries that conduct genetic sequencing for surveillance in this region should continue to work with the WHO CCs to produce high-quality genetic sequence data and upload those sequences to open-access databases. As the capacity for sequencing is strengthened in LA, there will be more real-time actionable information available to public health decision makers that will hopefully lead to improvement in the quality of seasonal influenza vaccines and earlier detection of the next pandemic virus.

## Supporting information

**S1 Table. Sequences from participating countries available in GISAID included in the study.**
(DOCX)

## Acknowledgments

The authors would like to thank Catherine Smith and Rebecca Garten from the WHO CC at the U.S. CDC for their mentorship in the development of this analysis and critical review of the manuscript, as well as Angel Rodriguez, Paulina Sosa, Paula Couto and Myrna Charles from the PAHO influenza team for the technical cooperation they provided to these countries to build their influenza surveillance systems.

## Author Contributions

**Conceptualization:** Juliana Almeida Leite, Paola Resende, Rakhee Palekar.

**Data curation:** Juliana Almeida Leite, Jenny Lara Araya, Gisela Badillo Barrera, Elsa Baumeister, Alfredo Bruno Caicedo, Leticia Coppola, Wyller Alencar de Mello, Domenica de Mora, Mirleide Cordeiro dos Santos, Rodrigo Fasce, Jorge Fernández, Natalia Goñi, Irma López Martínez, Jannet Otárola Mayhua, Fernando Motta, Maribel Carmen Huaringa Nuñez, Jenny Ojeda, María José Ortega, Erika Ospitia, Terezinha Maria de Paiva, Andrea Pontoriero, Hebleen Brenes Porras, Jose Alberto Diaz Quinonez, Viviana Ramas, Juliana Barbosa Ramírez, Katia Correa de Oliveira Santos, Marilda Mendonça Siqueira, Cynthia Vàzquez.

**Formal analysis:** Juliana Almeida Leite, Paola Resende, Jenny Lara Araya, Gisela Badillo Barrera, Elsa Baumeister, Alfredo Bruno Caicedo, Leticia Coppola, Wyller Alencar de Mello, Domenica de Mora, Mirleide Cordeiro dos Santos, Rodrigo Fasce, Jorge Fernández, Natalia Goñi, Irma López Martínez, Jannet Otárola Mayhua, Fernando Motta, Maribel Carmen Huaringa Nuñez, Jenny Ojeda, María José Ortega, Erika Ospitia, Terezinha Maria de Paiva, Andrea Pontoriero, Hebleen Brenes Porras, Jose Alberto Diaz Quinonez, Viviana Ramas, Juliana Barbosa Ramírez, Katia Correa de Oliveira Santos, Marilda Mendonça Siqueira, Cynthia Vàzquez, Rakhee Palekar.

**Methodology:** Juliana Almeida Leite, Rakhee Palekar.

**Project administration:** Rakhee Palekar.

**Software:** Paola Resende.

**Supervision:** Rakhee Palekar.

**Validation:** Juliana Almeida Leite, Paola Resende.

**Visualization:** Rakhee Palekar.

**Writing – original draft:** Juliana Almeida Leite, Paola Resende, Rakhee Palekar.

**Writing – review & editing:** Juliana Almeida Leite, Jenny Lara Araya, Gisela Badillo Barrera, Elsa Baumeister, Alfredo Bruno Caicedo, Leticia Coppola, Wyller Alencar de Mello, Domenica de Mora, Mirleide Cordeiro dos Santos, Rodrigo Fasce, Jorge Fernández, Natalia Goñi, Irma López Martínez, Jannet Otárola Mayhua, Fernando Motta, Maribel Carmen Huaringa Nuñez, Jenny Ojeda, María José Ortega, Erika Ospitia, Terezinha Maria de Paiva, Andrea Pontoriero, Hebleen Brenes Porras, Jose Alberto Diaz Quinonez, Viviana Ramas, Juliana

Barbosa Ramírez, Katia Correa de Oliveira Santos, Marilda Mendonça Siqueira, Cynthia Vàzquez, Rakhee Palekar.

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
