## [Decision Letter · Decision Letter 0]

10 Sep 2019

PONE-D-19-16505

Genetic evolution of influenza viruses among selected countries in Latin America, 2017–2018

PLOS ONE

Dear Dr Palekar,

Thank you for submitting your manuscript to PLOS ONE. After careful consideration, we feel that it has merit but does not fully meet PLOS ONE’s publication criteria as it currently stands. Therefore, we invite you to submit a revised version of the manuscript that addresses the points raised during the review process.

The manuscript has to undergo a language editing with the help of a native speaker and/or editing service. Samples were collected during a limited time period, please add a Discussion point as to how this could have affected the results.

In order to properly interpret the data from this manuscript, a comparison with global genetic diversity of the virus would be necessary.

We would appreciate receiving your revised manuscript by Oct 25 2019 11:59PM. To enhance the reproducibility of your results, we recommend that if applicable you deposit your laboratory protocols in protocols.io, where a protocol can be assigned its own identifier (DOI) such that it can be cited independently in the future. For instructions see: http://journals.plos.org/plosone/s/submission-guidelines#loc-laboratory-protocols

We look forward to receiving your revised manuscript.

Kind regards,

Peter Gyarmati

Academic Editor

PLOS ONE

Journal Requirements:

1. Please ensure that your manuscript meets PLOS ONE's style requirements, including those for file naming. The PLOS ONE style templates can be found athttp://www.journals.plos.org/plosone/s/file?id=wjVg/PLOSOne_formatting_sample_main_body.pdf and http://www.journals.plos.org/plosone/s/file?id=ba62/PLOSOne_formatting_sample_title_authors_affiliations.pdf

2. Please  change Fig.1, so that the names of the axes are clearly legible.

Additional Editor Comments (if provided):

Reviewers' comments:

Reviewer's Responses to Questions

**Comments to the Author**

1. Is the manuscript technically sound, and do the data support the conclusions?

Reviewer #1: Yes

Reviewer #2: Yes

Reviewer #3: Partly

2. Has the statistical analysis been performed appropriately and rigorously? 

Reviewer #1: Yes

Reviewer #2: Yes

Reviewer #3: N/A

3. Have the authors made all data underlying the findings in their manuscript fully available?

Reviewer #1: Yes

Reviewer #2: Yes

Reviewer #3: Yes

4. Is the manuscript presented in an intelligible fashion and written in standard English?

Reviewer #1: Yes

Reviewer #2: No

Reviewer #3: Yes

5. Review Comments to the Author

Reviewer #1: The manuscript analyzed influenza A/H1pdm09, A/H3, B/Victoria and B/Yamagata hemagglutinin sequences from clinical samples from 12 National Influenza Centers in ten countries with a collection date from epidemiologic week (EW) 18, 2017 through EW 43, 2018. These sequences were used for phylogenetic reconstruction. They reported that hemagglutinin sequences from the participating countries were highly concordant with the genetic groups of the influenza vaccine-recommended viruses for influenza A/H1pdm09 and influenza B. Since this study is helpful to allow public health decision makers to make informed decisions about prevention and control strategies, this study is relevant and would deserve publication. On the other hand, the quantity and analysis period of clinical samples and HA sequences of influenza virus is still limited, which influence the reliability of the study to some extent.

Reviewer #2: In this manuscript Juliana and coworkers construct the phylogeny consensus for the influenza virus on the basis of previously sequenced data from the data banks. They have studied the influenza A/H1pdm09, A/H3, B/Victoria and B/Yamagata HA sequences for the year 2017-2018. I think they have to expand the time duration as only for one year it can't be concluded that which specific strain is circulating in the area. In the manuscript there are several English mistakes. It need to read it properly and correct the English mistakes. In line 100 reference style is not correct. Please make sure all the references are correct and in appropriate order and style. Also make the sample numbers very clear it make the reader to confuse about the sample numbers for each specific strain of Influenza virus.

Reviewer #3: See attachment for full comments.

Overview:

The manuscript analyzes influenza virus evolution in Latin America using sequence data in the GISAID database. The authors identify a set of hemagglutinin sequences representing influenza A/H1pdm09, A/H3, and B generated from Latin American sequencing centers and from the US WHO Collaborating Center based on samples collected at Latin American sentinel sites from 2017 to 2018. They perform a phylogenetic analysis of these sequences, identify the substitutions that occur along the phylogeny, and calculate the frequencies of viral clades. They find that sequences from Latin America are broadly aligned with clade frequencies worldwide.

The manuscript addresses the important topic of influenza's genetic diversity in an under-surveilled part of the world. The analyses are appropriately conducted according to standard methods, although the phylogenetic methods themselves are not especially novel. I find that the manuscript could be strengthened by providing more precision about major conclusions regarding the evolution of viruses in Latin American and their clade frequencies relative to other areas of the world. Much of this work can be done through rewriting and clarifying the discussion, but some simple additional analyses would substantially strengthen the manuscript as well.

6. PLOS authors have the option to publish the peer review history of their article (what does this mean?). If published, this will include your full peer review and any attached files.

Reviewer #1: No

Reviewer #2: No

Reviewer #3: No

---

## [Author Response · Author response to Decision Letter 0]

8 Nov 2019

Dear Editors,

Thank you for your thoughtful review of our manuscript. Please find the responses to your comments below.

Sincerely,

Rakhee Palekar

General comments

Comment 1: The manuscript has to undergo language editing.

Response 1: This was done by a native English speaker, who is an influenza expert. 

Comment 2: Samples were collected during a limited time period, please add a Discussion point as to how this could have affected the results.

Response 2: We have added a sentence in the discussion.

Comment 3: In order to properly interpret the data from this manuscript, a comparison with global genetic diversity of the virus would be necessary.

Journal requirements

Comment. 4: Please ensure that your manuscript meets PLOS ONE's style requirements, including those for file naming. 

Response 4: We have done this. 

Comment 5: Please change Fig.1, so that the names of the axes are clearly legible.

Response 5: We have updated the figure. 

Reviewer specific comments

Reviewer #1: 

Comment 5: On the other hand, the quantity and analysis period of clinical samples and HA sequences of influenza virus is still limited, which influence the reliability of the study to some extent.

Response: Thank you for the comment. 

Reviewer #2

Comment 6: In line 100 reference style is not correct. Please make sure all the references are correct and in appropriate order and style. 

Response #6: The reference was corrected. 

Comment 7: Also make the sample numbers very clear it make the reader to confuse about the sample numbers for each specific strain of Influenza virus.

Comment 7: We have done this. 

Reviewer #3

Comment 8: The authors state in the discussion that viruses identified in Latin America resemble the viruses that are detected globally (lines 353-355). Although the authors compare the genetic groups that they identify with the vaccine strains selected in different years, they do not provide sufficient analyses to show to what extent the viruses sequenced in Latin America resemble global genetic diversity. To address this important and central question of geographic distributions, it would be helpful to do some of the following, in declining order of difficulty and importance: 

1. Provide a plot of global clade frequencies to parallel the plots of clade frequencies in Latin America in Figure 6. Generating these clade frequencies directly from global sequence data may be laborious, but the authors may be able to reproduce figures from other papers or to discuss clade frequencies in Latin America in comparison to those reported globally on sites like nextstrain.org. 

Response: We have added Figure 7 showing the global clades frequencies. The global findings align with the study results.

2. Provide a global phylogeny of influenza that includes some sequences from Latin America (subsampling the dataset would likely be necessary to conduct a more easily interpretable analysis), and label the sequences from Latin America. If these sequences are dispersed through the global tree, then this finding would strengthen the authors' argument that viruses in Latin America resemble global genetic diversity. 

Response: Based on the global distribution of the phylogenetic groups and clades, the phylogenetic inferences obtained for Latin America resembles the global genetic diversity. Since the main focus of the study was the Latin America countries, new phylogenetics trees were not added to avoid overloading the paper. 

3. Various other studies have addressed the geographic distribution of influenza, though not necessarily with respect to Latin America in particular (one example: https://www.ncbi.nlm.nih.gov/pubmed/26053121). The authors should cite more of this relevant literature to provide additional context for their conclusions. 

Response: We have added additional relevant literature and more points to the discussion and conclusions. 

Comment 9: The authors write in the discussion that "the viruses that circulated in these countries during the early part of the 2017 Southern Hemisphere influenza season evolved and changed as compared to those that circulated at the end of the 2018 Southern Hemisphere season" (lines 350-353). This statement is imprecise and could benefit from additional clarification. Evolution could refer to the accumulation of neutral mutations, antigenic drift, competition between clades carrying distinct antigenic mutations, and many other phenomena. In the rest of their study, the authors already identified some of the specific molecular changes that occurred, and while their analyses are not powered to identify the particular evolutionary forces at work, they could make this part of the discussion more detailed and precise. 

Response 9: We have updated this language. 

Comment 10: The authors should provide the standard acknowledgements table required for use of GISAID sequences. 

Response 10: A new supplemental table was provided in the GISAID standard acknowledgement table format.

Comment 11: axis labels on Figure 1 are upside down. 

Response 11: The axis labels were fixed. 

Comment 12: Sequence names in Figures 2-5 are mostly illegible, and the authors might consider replacing each sequence name with a colored dot representing time of collection instead. 

Comment 12: The sequences names were maintained in order to show the country of origin of the sequences.

Comment 13: lines 202-212: The number of HA sequences included in the final analysis (1395) is less than the numbers produced by the participating NICs and the number uploaded by the WHO CC at the US CDC (761 + 1169). Please clarify the reason for the discrepancy. Were the excluded sequences ones that had been passaged, or duplicates of other sequences, or was there some other reason? 

Response 13: Only sequences obtained from original clinical samples were included in the analysis; sequences obtained from multiple virus-passages, incomplete HA sequences, or HA sequences with mismatches/gaps were excluded (lines 187 to 190). This explains the “discrepancy.” We added a line in the Results to clarify this.

Comment 14: When defining clades and subclades, as in lines 227 and 232-233, for example, please clarify whether and when the clades being defined are part of a standard nomenclature. 

Response 14: We have reviewed this. 

Comment 15: in lines 365, 375, 382, 391, and 410, the authors write that about the "reduction" of a virus with ferret anti-sera raised against particular viral strains. Perhaps the authors should consider using the more common and clearer term "inhibition" or "hemagglutination inhibition" instead. 

Response 15: We have made this change in the Discussion.

---

## [Decision Letter · Decision Letter 1]

27 Nov 2019

PONE-D-19-16505R1

Genetic evolution of influenza viruses among selected countries in Latin America, 2017–2018

PLOS ONE

Dear Dr Palekar,

Thank you for submitting your manuscript to PLOS ONE. After careful consideration, we feel that it has merit but does not fully meet PLOS ONE’s publication criteria as it currently stands. Therefore, we invite you to submit a revised version of the manuscript that addresses the points raised during the review process.

We would appreciate receiving your revised manuscript by Jan 11 2020 11:59PM. To enhance the reproducibility of your results, we recommend that if applicable you deposit your laboratory protocols in protocols.io, where a protocol can be assigned its own identifier (DOI) such that it can be cited independently in the future. For instructions see: http://journals.plos.org/plosone/s/submission-guidelines#loc-laboratory-protocols

We look forward to receiving your revised manuscript.

Kind regards,

Peter Gyarmati

Academic Editor

PLOS ONE

Reviewers' comments:

Reviewer's Responses to Questions

**Comments to the Author**

1. If the authors have adequately addressed your comments raised in a previous round of review and you feel that this manuscript is now acceptable for publication, you may indicate that here to bypass the “Comments to the Author” section, enter your conflict of interest statement in the “Confidential to Editor” section, and submit your "Accept" recommendation.

Reviewer #1: All comments have been addressed

Reviewer #2: All comments have been addressed

Reviewer #3: (No Response)

2. Is the manuscript technically sound, and do the data support the conclusions?

Reviewer #1: Yes

Reviewer #2: Yes

Reviewer #3: Yes

3. Has the statistical analysis been performed appropriately and rigorously? 

Reviewer #1: Yes

Reviewer #2: Yes

Reviewer #3: N/A

4. Have the authors made all data underlying the findings in their manuscript fully available?

Reviewer #1: Yes

Reviewer #2: Yes

Reviewer #3: Yes

5. Is the manuscript presented in an intelligible fashion and written in standard English?

Reviewer #1: Yes

Reviewer #2: Yes

Reviewer #3: Yes

6. Review Comments to the Author

Reviewer #1: (No Response)

Reviewer #2: (No Response)

Reviewer #3: The manuscript analyzes influenza virus evolution in Latin America using next-generation sequencing data. In particular, the authors study the extent to which viral genetic diversity in Latin American countries reflects broader, global patterns of genetic diversity. Since the previously submitted version of the manuscript, which I reviewed, the authors have improved the comparison of influenza clade frequencies in Latin America compared to the rest of the world by including a new Figure 7, generated from Nextstrain. I am generally satisfied that the authors have addressed my concerns about presenting a more explicit comparison of regional and global genetic diversity, but the analysis in Figure 7 contains a few remaining oddities that the authors need to correct.

Figure 7 shows global clade frequencies for the four influenza subtypes under study and is derived from the visualization interface at nextstrain.org. It is not clear to me why the y-axes of these plots end at 12% (they should run from 0-100%), or why the diversity of viruses appears to decline after about mid-2018. Both anomalies should be corrected by the authors. For example, the nextstrain.org interface shows that for B/Yamagata, the 172Q clade makes up nearly all viral diversity following 2017, but in Figure 7, the clade appears to decline in frequency until it makes up none of the viral diversity under analysis. It seems likely that the authors set a cutoff date that causes the odd appearance of the graph, but this visualization choice is confusing. It would be better for the authors to present all clade frequencies between 2017 and the present.

7. PLOS authors have the option to publish the peer review history of their article (what does this mean?). If published, this will include your full peer review and any attached files.

Reviewer #1: No

Reviewer #2: Yes: Mohsan Ullah

Reviewer #3: No

---

## [Author Response · Author response to Decision Letter 1]

2 Jan 2020

Dear Editors,

We have updated Figure 7 as per Reviewer #3's recommendations. Thank you for your thoughtful review of our work.

Warm regards

Rakhee

---

## [Editor Report · Decision Letter 2]

6 Jan 2020

Genetic evolution of influenza viruses among selected countries in Latin America, 2017–2018

PONE-D-19-16505R2

Dear Dr. Palekar,

We are pleased to inform you that your manuscript has been judged scientifically suitable for publication and will be formally accepted for publication once it complies with all outstanding technical requirements.

With kind regards,

Peter Gyarmati

Academic Editor

PLOS ONE
---

## [Editor Report · Acceptance letter]

2 Mar 2020

PONE-D-19-16505R2 

Genetic evolution of influenza viruses among selected countries in Latin America, 2017–2018 

Dear Dr. Palekar:

I am pleased to inform you that your manuscript has been deemed suitable for publication in PLOS ONE. Congratulations! Your manuscript is now with our production department. 

With kind regards,

on behalf of

Dr. Peter Gyarmati 

Academic Editor

PLOS ONE